# Identification and Confirmation of Virulence Factor Production from *Fusarium avenaceum*, a Causal Agent of Root Rot in Pulses

**DOI:** 10.3390/jof10120821

**Published:** 2024-11-26

**Authors:** Thomas E. Witte, Anne Hermans, Amanda Sproule, Carmen Hicks, Tala Talhouni, Danielle Schneiderman, Linda J. Harris, Anas Eranthodi, Nora A. Foroud, Syama Chatterton, David P. Overy

**Affiliations:** 1Ottawa Research & Development Centre, Agriculture & Agri-Food Canada, 960 Carling Ave., Ottawa, ON K1A 0C6, Canada; tom.witte@agr.gc.ca (T.E.W.); anne.hermans@agr.gc.ca (A.H.); amanda.sproule@agr.gc.ca (A.S.); carmen.hicks@agr.gc.ca (C.H.); tala.talhouni@agr.gc.ca (T.T.); danielle.schneiderman@agr.gc.ca (D.S.); linda.harris@agr.gc.ca (L.J.H.); 2Lethbridge Research & Development Centre, Agriculture and Agri-Food Canada, 5403–1st Avenue South, Lethbridge, AB T1J 4B1, Canada; anas.eranthrodi@agr.gc.ca (A.E.); nora.foroud@agr.gc.ca (N.A.F.); syama.chatterton@agr.gc.ca (S.C.)

**Keywords:** *Fusarium avenaceum*, root rot disease, pulse crops, secondary metabolites, mycotoxins, virulence factors

## Abstract

*Fusarium avenaceum* is an aggressive pathogen of pulse crops and a causal agent in root rot disease that negatively impacts Canadian agriculture. This study reports the results of a targeted metabolomics-based profiling of secondary metabolism in an 18-strain panel of *Fusarium avenaceum* cultured axenically in multiple media conditions, in addition to an in planta infection assay involving four strains inoculated on two pea cultivars. Multiple secondary metabolites with known roles as virulence factors were detected which have not been previously associated with *F. avenaceum*, including fungal decalin-containing diterpenoid pyrones (FDDPs), fusaoctaxins, sambutoxin and fusahexin, in addition to confirmation of previously reported secondary metabolites including enniatins, fusarins, chlamydosporols, JM-47 and others. Targeted genomic analysis of secondary metabolite biosynthetic gene clusters was used to confirm the presence/absence of the profiled secondary metabolites. The detection of secondary metabolites with diverse bioactivities is discussed in the context of virulence factor networks potentially coordinating the disruption of plant defenses during disease onset by this generalist plant pathogen.

## 1. Introduction

The agricultural practice of monoculture cropping can often lead to ecological instability, involving reoccurring pathogen epidemics and a high demand for inputs such as fertilizers and herbicides/pesticides [1,2]. Crop diversification through annual crop rotation is a strategy aimed at increasing agroecosystem resilience and to stabilize particularly vulnerable cropping systems to a changing climate [3]. Cereals and pulses are among the top three crop product sectors in Western Canadian prairie provinces (alongside oilseeds). For example, the province of Saskatchewan in 2023 produced over 18.6 million metric tons and exported $6.9 billion (CAD) worth of cereal grains, and as Canada’s leading pulse producer (dry bean, lentils and peas), the province also yielded 2.8 million metric tons of pulses, exporting $3.2 billion (CAD) worth in 2023 [4]. The inclusion of pulses into wheat crop rotations is an adopted agriculture practice in the semi-arid Canadian prairie region due to the relatively better performance of pulse crops under water-stressed environments and subsequent wheat cropping can benefit from the residual soil nitrogen and moisture content left behind from the pulse crops [3].

Both root rot in pulse crops and Fusarium head blight of cereals are important fungal diseases that negatively impact Canadian farmers. Root rot in pulses is caused by a complex of pathogens, including *Fusarium avenaceum*, which is an aggressive pathogen in all pulse crops [5]. Fusarium head blight in cereals is also caused by a complex of *Fusarium* pathogens, of which *F. graminearum* is commonly found in epidemic years. However, *F. avenaceum* is regularly detected in western Canadian field surveys of Fusarium head blight-infected cereals [6,7], and durum wheat in particular. In Western Canada, where durum wheat–lentil rotations are common in conservation/zero-tillage environments, the durum–lentil rotation seems to be exasperating the problem of *Fusarium* diseases in these important cash crops, as *F. avenaceum* is frequently isolated from pea and lentil roots following previous planting and harvest of durum wheat [8]. Root rot occurrence caused by *F. avenaceum*, as assessed from pulse disease and pathogen surveys, has steadily increased in the Canadian prairies over the last 10–15 years [9]. Root rot is currently the most significant threat to the pulse industry in Saskatchewan and is responsible for causing up to 70–100% yield loss in severely infested fields; few effective control measures are currently available to reduce the severe impacts of this disease [9].

Understanding the infection strategies employed by fungal pathogens to infect their crop hosts is especially relevant in order to develop mitigation practices to control/prevent disease outbreaks. Virulence factors are small molecules secreted by pathogens to promote colonization and cell-to-cell transmission within a host. In well-studied pathosystems such as *F. graminearum* in cereals, multiple secondary metabolite/mycotoxin virulence factors have been identified, including deoxynivalenol and other trichothecenes, fusaoctaxins, fungal decalin-containing diterpenoid pyrones (FDDPs), and gramillins, many of which play a role in disease promotion by confounding the plant immune response [10,11,12,13]. Although *F. avenaceum* is known as a pathogen of cereals and pulses, knowledge regarding pathogenicity/virulence factors produced by *F. avenaceum* that play a role in disease virulence is limited. *F. avenaceum* is widely known for its capacity to produce mycotoxins such as enniatins, which pose an important threat to human and animal health [14]; however, the role of enniatins in *F. avenaceum* virulence in plants is unclear. Pathogenicity studies using enniatin knockout strains found that enniatins are not essential for *F. avenaceum* pathogenicity in pea and wheat [15,16], suggesting that other pathogenicity/virulence factors produced by *F. avenaceum* are also involved in causing disease.

The *F. avenaceum* genome is composed of 8 core chromosomes with over 60 secondary metabolite biosynthetic gene clusters [17,18]. Although *F. avenaceum* has the requisite biosynthetic gene clusters to make a vast array of secondary metabolites, only a relative limited number of secondary metabolite products have been empirically determined to be produced by *F. avenaceum*, particularly enniatins, moniliformin, fusarins, antibiotic Y, chlamydosporol, aurofusarin (and biosynthetic intermediates such as rubrofusarin and fuscofusarin), fusaristatins, JM-47, 2-amino-14,16-dimethyloctadecan-3-ol, acuminatopyrone, chrysogine, butanolide, ferricrocin and malonichrome [17]. Our current lack of knowledge regarding secondary metabolite virulence factors associated with *F. avenaceum* needs to be addressed before advances towards informed disease mitigation strategies can be actualized. By using information regarding known virulence factors derived from more well-characterized *Fusarium* pathosystems, the following research study used a targeted metabolomics and genome-mining approach to characterize the secondary metabolite potential of various *F. avenaceum* strains. Confirmation of in planta virulence factor expression was carried out using selected *F. avenaceum* strains during root rot challenge in two different pea cultivars, providing insights into potential mechanisms of *F. avenaceum* virulence via the coordinated expression of virulence factors.

## 2. Materials and Methods

### 2.1. Source of Isolates

All strains were isolated from cereal samples harvested in 2010/2011 (Canadian Grain Commission, Winnipeg, MB, Canada) and deposited in the Canadian Collection of Fungal Cultures (AAFC, Ottawa, ON, Canada). Strains have been given DAOMC culture collection accession IDs; see Appendix A. Species identification was confirmed by comparison of whole-genome sequences to FaLH03, FaLH27 and Fa05001, as previously published [17]. Enniatin synthase 1 (*Esyn1*) deletion mutants FaLH27Δ*esyn1_2* and FaLH27Δ*esyn1_8* are mutant strains derived from the FaLH27 parental line, in which *Esyn1* was disrupted. Neither strain produces enniatins. For details, see Eranthodi et al., 2020 [15].

### 2.2. Axenic Culture Conditions

The 18 *F. avenaceum* strains were inoculated into slants containing 15 mL of liquid media, placed at a 15° incline from the horizontal and grown at 25 °C in the dark. Four liquid media conditions (“CYA”, “MMK2”, “YES” and YES with sea salts added, or “YESIO”) were used, and strains were inoculated in a single replicate. Media recipes and growth conditions followed previously published procedures and media recipes utilized in a metabolomic analysis of *F. poae* strains [19]. In brief, the inoculated slants were incubated for 14 days, after which the mycelial mats were removed and frozen separately from the broth for independent extractions. Extractions were performed using ethyl acetate, by immersion of manually crushed mycelium samples, or as a liquid–liquid extraction of the broth samples, followed by shaking for 1 h. The extraction solvents were transferred to weighed boro-silicate scintillation vials, dried under vacuum and reweighed before being reconstituted in methanol at a concentration of 500 µg/mL for injection into the UPLC-HRMS system.

### 2.3. Pea Seedling Root Assays

This study utilizes a subset of the data obtained from experiments performed by Eranthodi et al., 2020, from pea seedling tissues extracted after 14 days of growth. A detailed account of the methods used for pea root infection, standardization and tissue preparation for metabolomics assaying is presented in Eranthodi et al., 2020 [15]. In brief, pea seeds were soaked in a macroconidial suspension, planted in vermiculite in isolated pots, and allowed to grow for 14 days. At harvest, the roots were washed and dried, and a 2 cm length of taproot that included the point of seed attachment was cut, frozen in liquid nitrogen and stored at −80 °C. The root tissue was freeze-dried and ground into a fine powder using a ball mill. To prepare the extracts, the samples were sonicated in a solvent mixture of water, acetonitrile, and acetic acid (20:79:1) for 4 min, followed by a 90 min incubation at room temperature on a nutating shaker. After clarification by centrifugation, a 150 µL aliquot was reserved for UPLC-HRMS analysis, utilizing the same equipment and chromatographic parameters as in the axenic-culturing metabolomics experiments.

### 2.4. Metabolic Profiling

All high-resolution mass spectrometry (HRMS) data were collected using an LTQ Orbitrap XL Hybrid Ion Trap mass spectrometer (Thermo Scientific, Waltham, MA, USA) coupled to a Dionex Ultimate 3000 ultra-high-performance liquid chromatography (UHPLC) system (Thermo Scientific). Chromatographic separation was performed using a Kinetex C18 column (50 mm × 2.1 mm, 1.7 µm; Phenomenex, Torrance, CA, USA) maintained at 30 °C and a flow rate of 0.350 mL/min. The mobile phase consisted of water with 0.1% formic acid (A) and acetonitrile containing 0.1% formic acid (B). The optimized 15 min mobile gradient consisted of mobile phase A, which was maintained at 95% for 0.5 min, before increasing to 95% solvent B over 4 min, and maintained at 95% solvent B for 3.5 min. The column was allowed to equilibrate for an additional 7 min with 95% solvent A. HRMS analysis parameters were as follows for *m*/*z* 100–2000 and resolution 30,000 in ESI^+^ mode: sheath gas (40), auxiliary gas (5), sweep gas (2), source voltage (4.0 kV), capillary temperature (320 °C), capillary voltage (35 V), and tube lens (100 V).

All “.RAW” data files, including samples, methanol blanks (run after every sixth sample), and medium controls, were processed using MZMine v2.51 (Cell Unit, Okinawa Institute of Science and Technology (OIST), Onna, Okinawa, Japan). Mass detection was carried out with a noise cut-off level of 1.0 × 10^3^. The ADAP algorithm for chromatograph building was used with the minimum group size set to 5 scans, the group intensity threshold set to 5.0 × 10^4^, and the minimum highest intensity set to 1.0 × 10^5^. Chromatographs were deconvoluted using the local minimum search with the chromatographic threshold set to 15.0%, the search minimum in the RT range set to 0.05 min, the minimum relative height set to 10.0%, the minimum absolute height set to 1.0 × 10^5^, the minimum ratio of peak top/edge set to 1.2 and the peak duration ranging from 0 to 2.00 min. Isotopes were grouped if they had a monotonic shape and a maximum charge of 2, with *m*/*z* tolerance set to 5 ppm and retention time tolerance set to 0.05 min. Mass features matching known or predicted secondary metabolite retention times (within 0.065 min) and *m*/*z* (within 5 ppm) were identified using the “custom database search”, supplied with a list of protonated and sodiated compound adduct *m*/*z* and retention times. All unidentified features were filtered from the data. The remaining, annotated features were aligned across samples using the Join Alignment function (with an *m*/*z* tolerance of 5.0 ppm and an RT tolerance of 0.05 min with a 2:1 weight for *m*/*z* vs. RT). Gaps in the dataset where variables fell below the chromatograph deconvolution detection threshold were backfilled using the gap-filling algorithm (using a *m*/*z* tolerance of 5 ppm and RT tolerance of 0.05 min). The data were then exported as .csv for further processing.

False positive peaks and peaks associated with media components were reduced/removed by subtracting maximum peak height intensities detected in the methanol blank samples run and uninoculated media samples from all other samples, using custom R scripts. The resulting dataset was thus “corrected” for the presence of column carryover by certain molecules, in particular enniatins, which are known to linger in the UPLC system. The corrected data were further simplified by obtaining the maximum peak height for each mass feature across all replicates of all media conditions for each strain, where applicable, to generate a representative secondary metabolite phenotype. Mass feature heights were converted to Log10 and visualized in a heatmap using the R package complexheatmap.

### 2.5. Metabolomic Annotation

Mass feature retention times and MS/MS fragmentation data were compared to purified standards for antibiotic Y (“lateropyrone”, Cayman chemical, Michigan, MI, USA), rubrofusarin (Sigma, St. Louis, MO, USA), enniatins A, A1, B and B1 (Sigma), FDDPs A-E (FDDPs provided by Dr. Tsukada’s lab at the University of Tokyo) and chlamydosporols (purified in-house as a mixture of two epimers). All others were annotated by comparison of exact *m*/*z* (<5 ppm), published UV absorbance spectra, and/or analysis of MS/MS fragmentation patterns. For the MS/MS data acquisition, spectra were generated using a Thermo qExactive mass spectrometer at the John H. Holmes Mass Spectrometry Facility at the University of Ottawa, using all parameters as published previously [19]. MS/MS spectral matching to MS/MS databases (MS-DIAL) [20] or rules-based filtering of structure libraries (MS-FINDER) [21] was used to support annotations where possible.

### 2.6. Genomic DNA Isolation, Sequencing and Assembly

Strains were grown for three days on glucose–yeast extract–peptone liquid media, after which the mycelium was freeze-dried. Genomic DNA was extracted using the Nucleon Phytopure genomic DNA extraction kit (GE Healthcare, Montreal, QC, Canada) and used for library preparation. The gDNA was mechanically sheared to 300 bp insert using the Covaris LE220 instrument (Covaris, Woburn, MA, USA). The inserts were used as a template to construct PCR-free Libraries with an NxSeq AmpFREE Low DNA Library kit (Lucigen, Middleton, WI, USA) and TruSeq CD dual indices (Illumina, San Diego, CA, USA) according to Lucigen’s Library protocol. Indexed libraries were pooled, and sequencing was carried on a NextSeq500/550 (Illumina) using a 2 × 150 bp NextSeq High Output Reagent Kit (Illumina) according to the manufacturer’s recommendations in order to obtain the paired-end reads. The paired-end reads were trimmed and corrected using fastp with the following flags: “-q 20 -l 50 –cut_front –cut_tail –correction –trim_poly_g”. Assemblies were produced using SPAdes with custom K-mers of lengths 21, 33, 55, 77 and 99. For assembly quality statistics, see Appendix A.

### 2.7. Gene Prediction and Annotation

Gene prediction was performed de novo using Funannotate v1.8.14 [22] on repeat-masked assemblies. Repeat masking was performed by first modeling transposable element families using RepeatModeler2, and then masking by supplying the de novo libraries to RepeatMasker. Funannotate was then used to predict gene models by comparing the outputs of Genemark-SH and Augustus was trained using *Fusarium graminearum* pretrained parameters, with SNAP and glimmerHMM weights set to zero. Gene models were refined where possible by comparing assembly regions to known BGCs associated with secondary metabolites of interest, using the “live annotate” feature in Geneious v2022.2. PKS and NRPS clade names were adopted from precedents in the literature [23,24] (Brown et al., 2022, Hansen et al., 2015).

## 3. Results and Discussion

### 3.1. Secondary Metabolite Profiling of Fusarium avenaceum Strains

The UPLC-HRMS profiles generated from in vitro culture extracts confirm that *Fusarium avenaceum* is a prodigious producer of small molecules (Figure 1). Targeted metabolomic profiling of 18 strains (Appendix A) cultured on diverse media conditions resulted in the detection of 31 secondary metabolites whose core structural scaffolds are biosynthesized by various classes of enzymes, including polyketide synthases (PKSs), non-ribosomal peptide synthetases (NRPSs), and “hybrid” biosynthetic gene clusters (BGCs) associated with multiple core scaffold synthases/synthetases. Secondary metabolites produced by PKS-associated BGCs included 2-amino-14,16-dimethyloctadecan-3-ol or AOD (**1**), antibiotic Y (**2**), aurofusarin (**3**), biosynthetic intermediates fuscofusarin (**4**) and rubrofusarin (**5**), acuminatopyrone (**6**), and chlamydosporols (**7**), which were detected as a mixture of epimers [25,26], along with the biosynthetic intermediate chlamydospordiol (**8**). Secondary metabolites produced by NRPS-associated BGCs included chrysogine (**9**), enniatin A (**10**), enniatin A1 (**11**), enniatin B (**12**), enniatin B1 (**13**), fusahexin (**14**), fusaoctaxin A (**15**), fusaoctaxin B (**16**), fusatrixin A (**17**), and fusapentaxin A (**18**). Secondary metabolites produced by “hybrid” BGCs included fungal decalin-containing diterpenoid pyrone (FDDP)-D (**19**), FDDP-E (**20**), fusarin A (**21**), fusarin C (**22**), fusarin PM (**23**), fusaristatin A (**24**), JM-47 (**25**), sambutoxin (**26**) and desmethyl-sambutoxin (**27**). Acetamido-butenolide and moniliformin are small, highly polar molecules associated with *F. avenaceum* metabolism that are not well retained on reversed-phase chromatography [19,27] and therefore were not detected in either in vitro culture extracts or in infected pea tissues, as the chromatographic parameters used were not optimized for the detection of small, highly polar molecules. Additionally, three fusaoctaxin analogs, one fusahexin analog and one putative FDDP shunt product were predicted based on their chemical formulas but have not yet been structurally elucidated. Associated mass spectrometry data are summarized in Appendix A with corresponding MS/MS mirrorplots and UV spectra in Appendix A.

Secondary metabolite profiles differed between strains (Figure 2). Mass features representing secondary metabolites that were consistently detected from all strains included AOD, antibiotic Y, enniatins (excepting two enniatin synthase knockouts included in the analysis), aurofusarin (and biosynthetic intermediates), fusahexin, JM-47, and sambutoxin. Differential detection of other mass features representing several secondary metabolites exhibited strain specificity, including acuminatopyrone, chlamydosporols, FDDPs, fusaristatin A, and fusarins. Some mass features displayed notable contrasts in relative abundance when observing UPLC-HRMS profiles of FaLH03 and FaLH27 cultured axenically and in planta. Fusaoctaxins and related analogs were inconsistently detected from extracts derived from axenic culturing (with the exception of the two enniatin synthase knockout strains—where the absence of enniatin production enabled detection of low levels of fusaoctaxin-associated mass features), but various fusaoctaxins were detected at high relative abundance from extracts of infected pea tissues. The inverse pattern was detected for the fusarins, where production was consistently observed from axenic cultures of producing strains but was largely absent from infected pea root tissue extracts.

### 3.2. Biosynthetic Gene Cluster Profiles Support Metabolomics Analysis

To further investigate the inconsistencies in secondary metabolite production observed from the various *F. avenaceum* strains, a comparative genomic approach was taken to scrutinize the architecture of associated BGCs between producing and non-producing strains (Figure 3A). The FDDP BGC [28] was found to be partially or wholly missing from the genomes of non-producing strains FaLH05, FaLH15, FaLH36 and FaLH37 (Figure 3B). Many of the genome assemblies encoded canonical telomeric repeats (TTAGGG) proximal to the FDDP BGC, suggesting the FDDP BGC is located near a telomere and that the FDDP cluster degradations in non-producing strains are associated with genomic instability of the telomeric region. The fusarin BGC [29] was predicted to be disrupted in the genomes of FaLH37 and FaLH39 by the presence of premature stop codons in the *PKS10* coding sequence, with FaLH39 also showing a loss of most of the BGC tailoring enzymes (Figure 3C). The fusaristatin A BGC [30], associated with *PKS6* and *NRPS7*, was not detected in the genome of the non-producing strain FaLH18; however, a different BGC associated with *PKS73* and *NRPS23* was detected in its place, suggesting that an unusual BGC “replacement” has occurred in this strain (Figure 3D). Interestingly, FaLH18 was the only strain from this set of *F. avenaceum* isolates in which the *PKS73*/*NRPS23* BGC was detected. The *PKS73*/*NRPS23* BGC is unassociated with a molecular product.

Lastly, chlamydosporol, chlamydospordiol and acuminatopyrone were present in extracts from all strains except FaLH18, FaLH27, FaLH32 and FaLH38. The co-occurrence of these molecules supports a common biosynthetic pathway for this molecular family, as has been suggested elsewhere [31]. Neither chlamydosporols nor acuminatopyrone production have been linked to a specific BGC, but based on their molecular structures they are most likely the products of a single, reducing Type 1 polyketide synthase (R-PKS). From the genome comparison of the various *F. avenaceum* strains, a BGC containing an R-PKS was predicted to be disrupted in all chlamydosporol/acuminatopyrone non-producing strains and intact in all producing strains (Figure 3E). The R-PKS gene is homologous with that of *PKS44* that was loosely associated with solanapyrone biosynthesis based on the BGC synteny match to the solanapyrone BGC in *Alternaria solani* [32]. The genome of FaLH18 was missing five out of six genes associated with the *PKS44* cluster, whereas the genomes of other non-producing strains FaLH27, FaLH32, and FaLH38 were predicted to have premature stop codons in the *PKS44* sequence. Based on this circumstantial evidence, and on the structural similarity between chlamydosporols and solanapyrones, a reasonable hypothesis can be proposed that the *PKS44* BGC produces chlamydosporols and acuminatopyrone in *F. avenaceum.*

### 3.3. Proposed Secondary Metabolite Virulence Factors Produced by F. avenaceum

Host plant infection by *F. avenaceum* is likely to involve the expression of a network of virulence factors rather than being driven by a single factor. Observations of *F. graminearum* during wheat infection have revealed that the plant immune response to pathogen ingress triggers early signaling responses including Ca^2+^ bursts, membrane depolarization, and ROS bursts, as well as late responses including transcriptional reprogramming and callose deposition to restrict pathogen movement [33]. Coregulation of virulence factors could enable *F. avenaceum* to bypass the plant immune system by deploying factors with complementary biological activities. Similarities in secondary metabolite production between *F. avenaceum* and *F. graminearum* supports the hypothesis that *F. avenaceum* metabolites have a cumulative impact during disease onset in root rot of pulses.

#### 3.3.1. Enniatins

Among the secondary metabolites produced by *F. avenaceum*, the enniatins have historically garnered the most research attention. UPLC-HRMS analysis of the *F. avenaceum* extracts from in vitro culturing experiments indicated that enniatins were the most abundant signals detected from all strains except in the two enniatin synthase knockout mutants derived from the FaLH27 parental line. Targeted enniatin profiling focused on enniatins A, A1, B and B1, although many more enniatin analogs are known to be produced by *F. avenaceum* [34] and may have been present as minor analogs. Of the enniatins observed from UPLC-HRMS profiling, enniatin B had the greatest relative abundance in terms of mass feature detection, from both in vitro culture extracts and from extracts of infected pea seedling tissue. Enniatins A, A1, B, and B1 produced from *F. avenaceum* were also the most frequently detected enniatins from silage and inoculated maize in prior studies [34,35].

There is some evidence to suggest a role of enniatins as virulence factors in studied plant pathosystems, but enniatin production does not appear to be essential for *F. avenaceum* disease in all plants. For example, exogenous application of enniatins inhibited seminal root elongation in 6-day-old germinated wheat seeds [36] and promoted tissue necrosis in sliced potato tubers [37]. Loss of enniatin production through deletion of the enniatin synthase gene led to a reduction in potato tuber necrosis by *F. avenaceum* deletion mutants as compared to a wild type [15,38]; however, loss of enniatin production was not found to impact an *F. avenaceum* challenge against durum wheat or pea seedlings [15] and was reported as a modest possible contributor to tissue-specific infection of common bread wheat [16].

Enniatins are cyclohexadepsipeptides that form potassium-selective, dimeric ionophore-ion complexes in biological membranes and facilitate passive K^+^ transport, disturbing physiological ionic balance and pH within cellular compartments, thereby affecting cellular homeostasis [14,39,40]. Destabilization of K^+^ homeostasis in potato tubers likely contributed to the necrosis phenotype associated with enniatins, as a deficiency of K^+^ flux is linked with increased tissue blackening in potato tubers [41]. Enniatins are also modulators of reticular Ca^2+^ channels associated with the mitochondria, endoplasmic reticulum and the cellular membrane, impacting cellular Ca^2+^ flux in mammalian cell models [42]. Ca^2+^ flux plays an essential role in pathogen-triggered plant immune responses; however, before assumptions can be made, further research is needed to extrapolate the impact of enniatins on Ca^2+^ flux in plant models. Enniatins are also known to impact cellular efflux of toxins via interaction with the yeast ATP-binding cassette transporter PDR5 [43]; homologs of PDR-encoding genes, including PDR5 gene homologs, are commonly found in plant genomes [44]. Repression of PDR5 toxin export could exacerbate disease pressure by increasing exposure of plant host cells to other virulence factors produced by *F. avenaceum* and other FHB or root rot pathogen complexes. Virulence factor efflux inhibition by enniatins could explain the reported increase in observed disease symptoms in peas co-contaminated with *F. avenaceum* and other root rot pathogens [8].

#### 3.3.2. FDDPs

Fungal decalin-containing diterpenoid pyrones (FDDPs) D and E were detected from in vitro cultures from all but four of the *F. avenaceum* strains, and were also present in diseased pea seedling tissue (Figure 2); this is the first report of FDDP production from *F. avenaceum*. Additionally, a mass feature was detected which matched the *m*/*z* of FDDP-B but did not elute at the same retention time as an FDDP-B standard, and was therefore annotated as a putative FDDP shunt product similar to colletochin [45], for which no reference standard was available. Production of FDDPs in other Fusaria appears to be non-constitutive and is associated with plant disease, as in vitro FDDP production by *F. graminearum* was only observed in a conserved number of culture medium formulations [13], whereas expression of the FDDP biosynthetic gene cluster expression was upregulated in *F. graminearum* infection cushions [46] and in planta production was consistently observed in wheat, barley and maize pathogenicity challenges [13,47,48]. The FDDPs are structurally similar to higginsianin B, a virulence factor produced by the plant pathogen *Colletotrichum higginsianum*, a causal agent of anthracnose disease of Brassicaceae [49]. Higginsianin B disrupts host–pathogen recognition and defense signaling during pathogen ingress by inhibiting proteolytic activity by the 26S proteosome, which is needed to process JASMONATE ZIM DOMAIN (JAZ)-containing proteins that play a role in the regulation of the jasmonic acid signaling pathway [49,50]. The terpenoid portion of higginsianin B is the ligand attributed to 26S proteosome inhibition and is identical to that of FDDP-D and FDDP-E, both in terms of molecular structure and stereochemistry [13]. Disruption of the terpene synthase gene of the FDDP BGC in *F. graminearum* resulted in a reduced disease phenotype in wheat heads, but not a complete arrest of pathogen dissemination, highlighting the fact that multiple virulence factors play a role in facilitating disease [13]. Although FDDP production can be hypothesized to contribute to *F. avenaceum* virulence in pea seedling root rot, this hypothesis requires further empirical validation.

#### 3.3.3. Fusaoctaxins

Results from the metabolomics data showed that fusaoctaxins A and B were detected at very low or trace levels only from in vitro culture extracts of the two *F. avenaceum* enniatin synthetase gene deletion mutants (*Fa*LH27Δ*esyn1_2* and *Fa*LH27Δ*esyn1_8*). Contrastingly, the metabolomes derived from infected pea tissue extracts showed that both fusaoctaxin forms were abundantly detected when peas were inoculated with FaLH03 and FaLH27 and also with the knockout mutants *Fa*LH27Δ*esyn1_2* and *Fa*LH27Δ*esyn1_8*. Fusaoctaxin analog-associated mass features, which have been detected previously from *F. graminearum* but remain uncharacterized at a structural level and are arbitrarily labeled fusaoctaxin analogs 2, 3 and 4 [51], were similarly detected at high levels in infected plant tissue as compared to low or no detection from in vitro experiments. The presence of the complete, eight-gene fusaoctaxin biosynthetic gene cluster [11] in all *F. avenaceum* genome assemblies confirmed metabolomics observations that all of the *F. avenaceum* strains were capable of fusaoctaxin production. Cleaved fusaoctaxin products fusatrixin A and fusapentaxin A, reported to be produced during fusaoctaxin export [52], were also detected with varying consistency from infected pea tissues: while the fusatrixin A-associated mass feature was detected in extracts from all strains, fusapentaxin A was only detected from a subset of strains, and fusatetraxin A was not detected. While little is known about the mode of action of fusaoctaxins in plant model systems, they have been reported as virulence factors involved in the cell-to-cell transmission of *F. graminearum* in wheat coleoptiles and have been linked to the inhibition of callose deposition during the plant defense response [11]. Furthermore, the expression of the fusaoctaxin biosynthetic gene cluster has been linked to co-expression of the FDDPs in *F. graminearum* during the formation of infection cushions [46] and was detected 64–96 h after infection of barley and wheat kernels [53].

#### 3.3.4. JM-47

The cyclic lipopeptide JM-47 was also detected in culture extracts from all *F. avenaceum* strains profiled as well as from infected pea root tissues. All genomes contained the JM-47 biosynthetic gene cluster as proposed by Lysøe et al. (2014) [17]. The molecule JM-47 is structurally almost identical to HC-toxin produced by *Cochliobolus carbonum*, the causal agent of Northern corn leaf spot disease [54]. Both JM-47 and HC-toxin are cyclic tetrapetides containing three amino acid residues (Pro-Ala-Ala) bonded with an uncommon ten-carbon lipoamino acid with a terminal 8-oxo group (an α-hydroxy ketone in JM-47 and an epoxy ketone in HC-toxin). JM-47 is also structurally similar to the cyclic tetrapeptide apicidins produced by several plant pathogenic *Fusarium* spp. [55]. The apicidins, HC-toxin and several other synthetic structural analogs are known histone deacetylase (HDAC) inhibitors, suggesting an HDAC inhibitory property for JM-47 [56]. Reversible histone acetylation is implicated in *epi*-genetic regulated gene expression in numerous in planta biological processes, such as the switching of metabolic states during plant pathogenesis [54]. Non-competitive and reversible HDAC inhibitory activity of HC-toxins has been observed for Rpd3/Hda1 class HDACs and nucleolus-localized HD2 class HDACs in plants. In corn and other cereals, the epoxide moiety of HC-toxin is hypothesized to be responsible for the heightened virulence of *C. carbonum* strains in susceptible corn lines. Non-susceptible corn lines have evolved a NAD(P)H-dependent HC-toxin reductase to detoxify HC-toxin (likely due to the reduction of the epoxide), orthologs of which are also found in other cereal plants (barley, sorghum, oats), but interestingly, HC-toxin reductase orthologs have not been detected in dicots such as pulses [54]. Therefore, it is possible that the proposed HDAC inhibitory activity of the structural analog JM-47 could contribute to pathogenicity in pulses and other dicots.

#### 3.3.5. Moniliformin

Moniliformin is a known phytotoxic virulence factor, and while it is produced by *F. avenaceum* during infection of various cereal crops [57,58,59], it is not known whether moniliformin was produced during infection of pea roots in the work presented herein due to the incompatibility of the molecule with the chromatography–mass spectrometry methods employed. Moniliformin inhibits pyruvate and α-ketoglutarate oxidation, which is a critical biochemical process in metabolic pathways in animals and plants [60]. Several phytotoxic effects, such as reduced growth rates, chlorosis and necrosis following exposure of moniliformin, have been reported in different monocot and dicot plant studies involving wheat, maize, tobacco and duckweed (reviewed in Jestoi, 2008) [61]. To date, the biosynthetic gene cluster associated with moniliformin remains unknown.

#### 3.3.6. Sambutoxin

Sambutoxin was produced by all of the *F. avenaceum* strains under study and was observed in infected pea tissues. Within the genus *Fusarium*, reported production of sambutoxin from in vitro cultivation has been limited to strains of *F. sambucinum*, *F. oxysporum*, and a single strain of *F. semitectum*; sambutoxin was also associated with *F. sambucinum* and *F. oxysporum* rot of potato tubers [62]. The potential for sambutoxin to be produced by *F. avenaceum* was suggested by Brown et al. (2022) based on biosynthetic gene cluster similarity [23]; our current study is the first report of sambutoxin production from *F. avenaceum* and the in planta production of sambutoxin from infected pea seedlings. Little is known about the potential of sambutoxin as a virulence factor in plants, but the molecule has demonstrated potent biological activity in mammalian cells. In human breast cancer cell models, sambutoxin exposure resulted in a dose-dependent increase in cellular apoptosis that was induced through cellular ROS generation and subsequent activation of programmed cell death via the mitogen activated protein (MAP) kinase, c-Jun NH_2_-terminal kinase (JNK)-mediated, mitochondrial apoptosis pathway [63]. Sambutoxins could contribute to virulence in pea roots via a similar disruption of plant cellular functions.

#### 3.3.7. AOD

AOD was detected in all of the *F. avenaceum* strains grown in vitro and from pea tissue infected with FaLH03 or FaLH27 (Figure 2). This is consistent with what has previously been observed in *F. avenaceum*-infected wheat heads [64,65]. The molecule AOD is classed as a sphingosine-analog metabolite, due to its presumptive structural mimicry of sphingosine, an important structural and signaling molecule in eukaryotic membranes [64]. Well-studied sphingosine-analog metabolites include AAL toxin and fumonisins, which are produced by phytopathogenic fungi and derive their toxic activity to eukaryotic cells by inhibition of dihydroceramide production, an important building block for more complex sphingolipids [65]. Although sphingosine mimicry is potentially disruptive to all eukaryotic cells, to date no studies have investigated the effects of AOD on plant cells or on the timing of AOD production during plant infection.

#### 3.3.8. Fusahexin and Fusahexin Analogs

Fusahexin production was detected at low levels in all extracts of the *F. avenaceum* strains cultured in vitro, but fusahexin was not detected in infected pea tissue samples. A second mass feature showing a similar chemical formula (C_30_H_52_N_6_O_8_) corresponding to a neutral gain of H_2_O compared to fusahexin (C_30_H_50_N_6_O_7_) was annotated as “fusahexin analog” and was detected at higher relative abundance compared to fusahexin from all extracts including infected pea tissue. At this time, the structure of the fusahexin analog is not known, but given the mass difference of a water molecule, it is possibly a hydrolyzed form of fusahexin which has not yet been described. All *F. avenaceum* genomes encoded homologs of the two-gene NRPS4 BGC associated with fusahexin biosynthesis in *F. graminearum* [66]. At this time, although fusahexins have been observed in both pea and wheat pathogen challenges by *F. avenaceum* and *F. graminearum,* respectively, their involvement in pathogenicity remains unclear. Overexpression of the fusahexin biosynthetic gene cluster, rather than disruption of the gene cluster, was observed to reduce/limit kernel-to-kernel disease progression of *F. graminearum* in wheat heads [66].

#### 3.3.9. Other Metabolites Not Currently Associated with Pathogenicity or Virulence

Both in vitro and in planta metabolomes of the *F. avenaceum* strains were replete with secondary metabolites which so far lack sufficient evidence to be implicated in plant pathosystems as virulence factors. Fusarins A, C and PM were detected from all strains except FaLH37 and FaLH39; however, the fusarins were only detected at trace levels or not at all from inoculated pea root samples. Fusarins have been investigated as mycotoxins due to their reported mutagenic and mycoestrogenic properties [67], but their effects on plant host interactions, if any, remain unreported. The lipopeptide fusaristatin A was predominantly associated with the *F. avenaceum* mycelium extracts (with the exception of FaLH18), which is consistent with previously reported patterns of fusaristatin A production by *F. graminearum* [68], suggesting that it is not a secreted virulence factor. Relatively little is known about the potential biological role of fusaristatin A in plant pathosystems; interestingly, the loss of fusaristatin A production by *F. pseudograminearum* was associated with an increased growth rate as compared to the wild-type strain in a wheat crown rot infection assay [69]. All of the *F. avenaceum* strains cultured in vitro produced antibiotic Y; as the name implies, antibiotic Y (also known as avenacein Y and lateropyrone) is notable for its strong antibiotic effects [70,71]. Production of antibiotic Y was detected only at trace levels in infected pea plants, suggesting a non-functional role of this molecule as a potential virulence factor in root rot in pulses, which is consistent with previous studies reporting a lack of phytotoxicity of antibiotic Y in wheat and pea seedlings [72]. The pigment molecules chrysogine and aurofusarin and the biosynthetic intermediates fuscofusarin and rubrofusarin were consistently found in extracts from all *F. avenaceum* strains in both in vitro and in planta experiments. Both chrysogine and aurofusarin are rather common pigments that are produced by a variety of different fungi [71]; although the production of both chrysogine and aurofusarin has been extensively investigated, the lack of evidence of in planta biological effect suggests that these molecules are likely not virulence factors associated with plant disease. Acuminatopyrone, chlamydosporol and associated metabolites were produced by all but four of the *F. avenaceum* strains (FaLH18, FaLH27, FaLH32 and FaLH38). Although chlamydosporol is associated with cytotoxicity in mammalian cell lines and animal models [73], to date, there are no literature reports of involvement of chlamydosporol or acuminatopyrone in plant disease or cytotoxicity in plant assays. In seedling assays of the annual grass *Bromus tectorum*, exposure to different chlamydosporol analogs did not affect coleoptile or radicle development, but moderate inhibitory activity was observed for acuminatopyrone [74]. It is therefore unlikely that chlamydosporols are involved in the pathogenicity of *F. avenaceum* in pea.

## 4. Conclusions

This study represents the first confirmed report of fusaoctaxin, fusahexin, FDDP, and sambutoxin production by *Fusarium avenaceum*, significantly expanding our understanding of the species’ secondary metabolite profile. The diversity of *F. avenaceum* secondary metabolism profiled so far, with its associated range of potential biological activities in plant pathosystems, strongly suggests that host infection involves the coordinated expression of virulence factor networks rather than being driven by a single component. The co-regulation of these factors may enable *F. avenaceum* to bypass the plant immune system, deploying virulence mechanisms with complementary biological activities. Importantly, enniatins affect cellular homeostasis, modulate Ca^2+^ flux and impact cellular efflux of toxins—and their presence is therefore likely to increase the efficacy of other virulence factors produced by *F. avenaceum* and/or other pathogens in infected host plant cells, thereby exacerbating disease pressure from pathogen complexes. The reported toxicities of secondary metabolites detected in infected pea tissue in this study could therefore be augmented by their co-occurrence with enniatins, and each other: FDDPs likely disrupt host–pathogen recognition and defense signaling during pathogen ingress; JM-47 inhibits HDAC-associated control of metabolic states during plant pathogenesis; fusaoctaxins facilitate cell-to-cell transmission; AOD compromises membrane integrity and cell signaling processes by disrupting sphingolipid metabolism; and moniliformin inhibits key primary metabolic functions in eukaryotic cells, presaging reduced growth rates, chlorosis and necrosis. These complementary activities could enable *F. avenaceum* to overcome the redundancy of plant defenses, facilitating pathogen ingress, cell-to-cell transmission, and disease onset.

These hypotheses, particularly the coordinated action of multiple virulence factors, require further testing and are important concepts when considering how generalist pathogens such as *F. avenaceum* have evolved to infect diverse plant hosts and tissue types, such as pea roots or cereal heads. Gene-editing tools such as CRISPR could be employed to create multi-gene knockout mutants, providing insight into how these factors interact and contribute to pathogenicity. Additionally, purification of individual secondary metabolites for plant-based assays could help clarify their specific roles in disease progression. Preventing fungal epidemics and mycotoxin contamination remains a top priority for cereal and pulse producers. Gaining a deeper understanding of these virulence factors offers potential targets for developing new mitigation strategies, which are crucial for Canadian agriculture to ensure a safe food supply and value chain.

## Figures and Tables

**Figure 1 jof-10-00821-f001:**
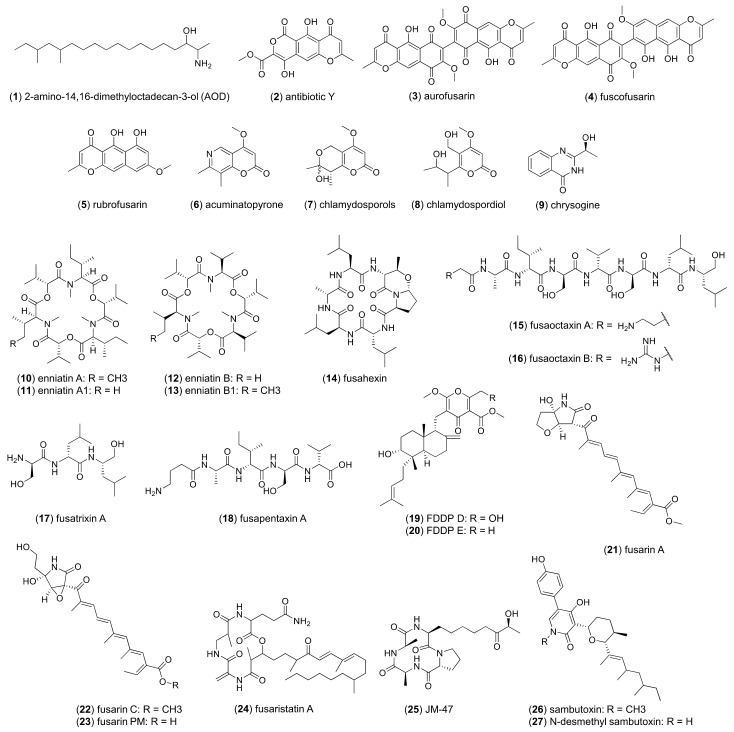
Molecular structures of *F. avenaceum* secondary metabolites produced from in vitro culturing and in planta pathogenicity challenge in pea seedlings.

**Figure 2 jof-10-00821-f002:**
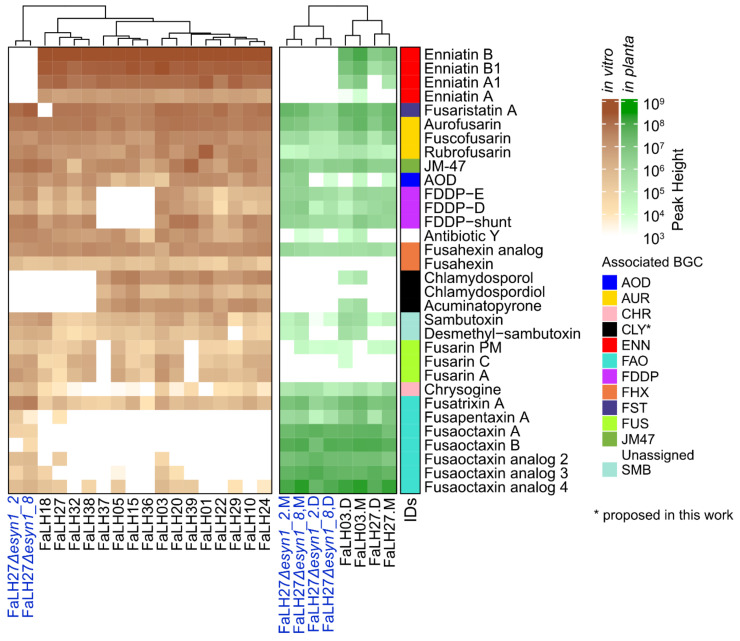
Targeted metabolomics analysis of 16 *Fusarium avenaceum* strains and two enniatin synthase (ENN) gene deletion mutants (blue strain IDs) generated from parental strain FaLH27. The heatmap on the left shows representative mass feature peak heights detected across all media conditions used in the experiment; the heatmap on the right (in green) shows peak heights detected from infected pea root tissue. In planta inoculations were performed on two pea cultivars: CDC Meadow (M) and CDC Dakota (D).

**Figure 3 jof-10-00821-f003:**
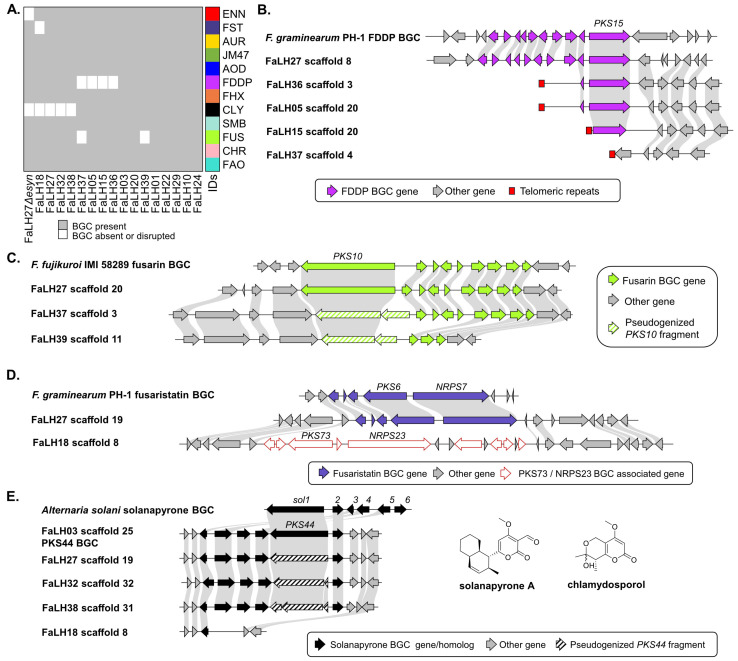
Annotated mycotoxin/metabolite biosynthetic gene clusters (BGCs) in *F. avenaceum* genomes. (**A**) Presence/absence heatmap of BGCs in all strains included in this study. (**B**) Syntenic comparison of *F. graminearum* FDDP BGC to one FDDP-producing strain and all non-producing *F. avenaceum* strains. (**C**) Syntenic comparison of the fusarin BGC from *F. fujikuroi* to one fusarin-producing strain and all non-producing *F. avenaceum* strains. (**D**) Syntenic comparison of the fusaristatin BGC from *F. graminearum* to a producing and a non-producing *F. avenaceum* strain. (**E**) Syntenic comparison of solanapyrone BGC from *Alternaria solani* to the *PKS44* BGC in one chlamydosporol-producing strain and all non-producing *F. avenaceum* strains.

## Data Availability

Access to all fungal strains used/generated in this research can be obtained through Agriculture & Agri-Food Canada via consultation with the corresponding author or by contacting the Canadian Collection of Fungal Cultures (AAFC, Ottawa, ON, Canada).

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
