# Peer review of "Identification and Confirmation of Virulence Factor Production from Fusarium avenaceum, a Causal Agent of Root Rot in Pulses"

_jof, 2024, doi:10.3390/jof10120821_

Round 1

Reviewer 1 Report

The report by Witte, et al. profiled 18 strains of Fusarium avenaceum (Fa) for the production of secondary metabolites implicated in virulence under a variety of media conditions in vitro and a subset of isolates in planta during infection of pea roots. The authors identified metabolites known to be produced by Fa, as well as several metabolites not previously reported to be biosynthesized by Fa. Confirmation of the mass spectrometry data was performed at multiple annotation levels with varying degrees of confidence (summarized in Table S2). The mass spec data was further validated using whole genome sequencing and targeted genomic analyses of biosynthetic gene clusters to support the production of the identified compounds, focusing on metabolites that exhibited strain-to-strain variability. These analyses supported the presence or absence of intact BGCs, which corroborated the metabolomic profiling. The authors discuss the relevance of the detected compounds in the context of their known and potential modes of action during plant infection. Overall, the manuscript is well-written and easy to follow. This is the first report on the production of fusaoctaxin, fusahexin, FDDP, and sambutoxin by Fa, and thus should be of interest to scientists working on Fusarium and other fungal plant pathogens. This expanded knowledge on the metabolic cocktails produced by Fa will further investigations into mechanisms of fungal virulence and the genetic basis for biosynthesis of specific metabolites. There are some relatively minor issues outlined below, but once resolved, I would recommend this manuscript to be published in JoF.

Major points:

1) Supplemental Figures S1-S7: The provided pdf has an issue with the mirror plots, and the ion peaks are not displayed on the plots, possibly due to issues when converted to pdf format. This issue makes it more difficult to evaluate these data. It is unclear which of the upper or lower plots represents the experimental data collected in this study or the expected fragmentation patterns from the library.

2) The authors should provide the sequencing and assembly statistics for the whole genome sequencing of the Fa isolates presented here, including measures of completeness. The raw data, assembled data, and genome annotations should also be made available at an appropriate repository, and referenced in a “Data Availability” statement.

L30: remove extra “.” at the end of the sentence

L100: “Source of isolates” section: provide information for the ESYN1 disruption mutants (FaLH27∆esyn1_2 and FaLH27∆esyn1_8) used in this study, including the appropriate reference.

L123: provide numbered citation, i.e. “Eranthodi et al. 2020 [15]”

L290: suggest replacing “population” with a different term/phrase, e.g. “from this set of F. avenaceum isolates”

L365-368: “Repression of PDR5 toxin export could increase exposure of plant host cells to other virulence factors produced by F. avenaceum and exacerbating disease pressure during pathogen challenge with FHB and root rot pathogen complexes involving virulence factors produced by other organisms.” – Suggest revising and breaking up this sentence for clarity. There are other examples of inordinately long sentences in the manuscript that could also be broken up into smaller sentences for clarity.

L378: “epigenetically regulated” is likely not the correct term here. Mechanisms of epigenetic regulation are typically associated with DNA methylation, histone modifications, chromatin remodeling, and RNAi. From the information cited, I understand that FDDP production is not constitutive, but this does not necessarily make it “epigenetically regulated”.

L542: “component” instead of “proponent”?

Supplemental Table S1: Define abbreviations for Province names.

Supplemental Figure S1C: The x-axis appears to be a typo and should be “wavelength (nm)”

Author Response

Reviewer 1:

Major Points:

Comment 1:  Supplemental Figures S1-S7: The provided pdf has an issue with the mirror plots, and the ion peaks are not displayed on the plots, possibly due to issues when converted to pdf format. This issue makes it more difficult to evaluate these data. It is unclear which of the upper or lower plots represents the experimental data collected in this study or the expected fragmentation patterns from the library.

Response 1: You are correct – the PDF export ruined the mirror plots!  Thank you for bringing this to our attention – we have exported all plots as png files and replaced the faulty plots.  Additionally, we have edited the captions to clarify interpretation of the mirror plots.

Comment 2: The authors should provide the sequencing and assembly statistics for the whole genome sequencing of the Fa isolates presented here, including measures of completeness.

Response 2: Associated sequencing and assembly statistics for the whole genome sequencing of the Fa isolates have been provided in detail in the supplementary information document.

Minor Comments:

Comment 3: L30: remove extra “.” at the end of the sentence

Response 3: Done.

Comment 4: L100: “Source of isolates” section: provide information for the ESYN1 disruption mutants (FaLH27∆esyn1_2 and FaLH27∆esyn1_8) used in this study, including the appropriate reference.

Response 4: Text added: Enniatin synthase 1 (Esyn1) deletion mutants FaLH27Δesyn1_2 and FaLH27Δesyn1_8 are mutant strains derived from the FaLH27 parental line, in which Esyn1 was disrupted. Neither strain produces enniatins. For details see Eranthodi et al. 2020 [15].

Comment 5: L123: provide numbered citation, i.e. “Eranthodi et al. 2020 [15]”

Response 5: Done.

Comment 6: L290: suggest replacing “population” with a different term/phrase, e.g. “from this set of F. avenaceum isolates”

Response 6: Replaced as suggested.

Comment 7: L365-368: “Repression of PDR5 toxin export could increase exposure of plant host cells to other virulence factors produced by F. avenaceum and exacerbating disease pressure during pathogen challenge with FHB and root rot pathogen complexes involving virulence factors produced by other organisms.” – Suggest revising and breaking up this sentence for clarity. There are other examples of inordinately long sentences in the manuscript that could also be broken up into smaller sentences for clarity.

Response 7: Agreed.  This sentence now reads “Repression of PDR5 toxin export could exacerbate disease pressure by increasing exposure of plant host cells to other virulence factors produced by F. avenaceum and other FHB or root rot pathogen complexes.”

Comment 8: L378: “epigenetically regulated” is likely not the correct term here. Mechanisms of epigenetic regulation are typically associated with DNA methylation, histone modifications, chromatin remodeling, and RNAi. From the information cited, I understand that FDDP production is not constitutive, but this does not necessarily make it “epigenetically regulated”.

Response 8: Agreed – there could be other mechanisms (eg. transcription factors) which influence expression.  We have revised this paragraph for clarity.

Comment 9: L542: “component” instead of “proponent”?

Response 9: Done.

Comment 10: Supplemental Table S1: Define abbreviations for Province names.

Response 10: Done.

Comment 11: Supplemental Figure S1C: The x-axis appears to be a typo and should be “wavelength (nm)”

Response 11: Good catch!  You are correct. X-axis label changed.

Reviewer 2 Report

This article is interesting and well written. It reports virulence factor production from Fusarium avenaceum. It contains important information to be published, but there are several points to be improved before publication.

1. Abstract. Please summarize and simplify the first three sentences.

2. Line 30. Please delete superfluous ..

3. Line 49. The name of species and the genus should be written in italics. Please check the whole text.

4. Line 123&125. Eranthodi et al. 2020. Please follow the specified reference format.

5. Reference part. Format inconsistency. For example, No.1-3 are obviously different from the rest. Please check them and correct.

6. Table S1. NA, SK… refer to ? Please add the meaning of the abbreviation.

7. Table S2. The format of the tables must be consistent.

Author Response

Reviewer 2:

This article is interesting and well written. It reports virulence factor production from Fusarium avenaceum. It contains important information to be published, but there are several points to be improved before publication.

Thank you for your kind comments and for reviewing this article.

Comment 1: Abstract. Please summarize and simplify the first three sentences.

Response 2: We have rewritten the first three sentences to remove superfluous points.  It is now one sentence instead of three.

Comment 2: Line 30. Please delete superfluous ‘.’.

Response 2: Done.

Comment 3. Line 49. The name of species and the genus should be written in italics. Please check the whole text.

Response 3: The reviewer is referring to our use of the term “Fusarium head blight”.  In this case, we are referring to the disease, not the genus.  Therefore, it requires no italicization.  This is standard for scientific writing in the field of Fusarium head blight research.  We have left the text unchanged.

Comment 4. Line 123&125. Eranthodi et al. 2020. Please follow the specified reference format.

Response 4: Done.

Comment 5. Reference part. Format inconsistency. For example, No.1-3 are obviously different from the rest. Please check them and correct.

Response 5: Agreed – references have been corrected.

Comment 6. Table S1. NA, SK… refer to ? Please add the meaning of the abbreviation.

Response 6: Done.

Comment 7. Table S2. The format of the tables must be consistent.

Response 7: We have adjusted the format of Table S1 to match Table S2.